# Dioctyl Phthalate-Modified Graphene Nanoplatelets: An Effective Additive for Enhanced Mechanical Properties of Natural Rubber

**DOI:** 10.3390/polym14132541

**Published:** 2022-06-22

**Authors:** Linh Nguyen Pham Duy, Chuong Bui, Liem Thanh Nguyen, Tung Huy Nguyen, Nguyen Thanh Tung, Duong Duc La

**Affiliations:** 1Center for Polymer Composite and Paper, School of Chemical Engineering, Hanoi University of Science and Technology, Hai Ba Trung, Hanoi 100000, Vietnam; linh.nguyenphamduy@hust.edu.vn (L.N.P.D.); chuong.bui@hust.edu.vn (C.B.); liem.nguyenthanh@hust.edu.vn (L.T.N.); tung.nguyenhuy@hust.edu.vn (T.H.N.); 2Institute of Materials Science and Graduate University of Science and Technology, Vietnam Academy of Science and Technology, 18-Hoang Quoc Viet, Hanoi 100000, Vietnam; tungnt@ims.vast.ac.vn; 3Institute of Chemistry and Materials, Hoang Sam, Nghia Do, Cau Giay, Hanoi 100000, Vietnam

**Keywords:** graphene nanoplatelets, modified graphene, rubber, dioctyl phthalate, nanocomposite

## Abstract

Graphene has been extensively considered an ideal additive to improve the mechanical properties of many composite materials, including rubbers, because of its novel strength, high surface area, and remarkable thermal and electron conductivity. However, the pristine graphene shows low dispersibility in the rubber matrix resulting in only slightly enhanced mechanical properties of the rubber composite. In this work, graphene nanoplatelets (GNPs) were modified with dioctyl phthalate (DOP) to improve the dispersibility of the graphene in the natural rubber (NR). The distribution of the DOP-modified GNPs in the NR matrix was investigated using scanning electron microscopy, X-ray diffraction, and Raman spectroscopy. The effect of the modified GNPs’ contents on the mechanical properties of the GNPs/NR composite was studied in detail. The results showed that the abrasion resistance of the graphene-reinforced rubber composite significantly improved by 10 times compared to that of the rubber without graphene (from 0.3 to 0.03 g/cycle without and with addition of the 0.3 phr modified GNPs). The addition of the modified GNPs also improved the shear and tensile strength of the rubber composite. The tensile strength and shear strength of the NR/GNPs composite with a GNPs loading of 0.3 phr were determined to be 23.63 MPa and 42.69 N/mm, respectively. Even the presence of the graphene reduced the other mechanical properties such as Shore hardness, elongation at break, and residual elongation; however, these reductions were negligible, which still makes the modified GNPs significant as an effective additive for the natural rubber in applications requiring high abrasion resistance.

## 1. Introduction

Rubber is considered one of the most commercially used polymers in every aspect of living and for industrial activities. Rubber precursors could be categorized into natural and synthetic rubber types (polybutadiene rubber, styrene-butadiene rubber, isobutylene isoprene rubber…) [1]. Among these, natural rubber (NR), which is low cost, highly flexible, has good mechanical properties, is found in abundance, especially in the South of Vietnam, is commonly utilized in many industrial products such as wheel cars, conveyors, and many consumer products [2,3]. One of the most used fields for NR rubber is the tire industry, which requires high wear resistance, tensile strength, and durability. Since NR has low abrasion resistance, it is necessary to compound it with fibers and fillers to improve the mechanical properties, especially its abrasion resistant capacity for tire application [4,5,6,7,8,9,10]. Many fillers such as carbon black, silica, clay, natural fiber, and biomass fillers have been commonly used to reinforce rubber [11,12]. However, such fillers need to be added at a high proportion to NR in order to enhance the mechanical properties (for example, the carbon black filler), resulting in high density and reducing several mechanical properties of the elastomeric material [13]. It is also well-perceived that the production of the carbon black filler emits a considerable amount of CO_2_ and pollutant wastes, which causes severe problems for the ecosystem and human health [14]. Recently, alternative nano-scale fillers have been extensively studied and are considered an ideal replacement for the conventional fillers in the rubber industry, because of their low adding proportion with a significant enhancement of the desired mechanical properties as well as their ability to minimize the side-effects on the other properties [15,16].

Since the first discovery by Andre Geim and Konstantin Novoselov in 2004, graphene has been extensively studied for many applications including, but not limited to, sensing, composites, catalyst, adsorption, supporting materials, and electronic devices due to their novel chemical and physical properties [17,18,19,20,21,22,23]. Among these, the use of graphene as a nanofiller in composites is one of the most promising application fields, which extensively attracts scientists worldwide [24,25,26]. Graphene nanofillers could improve many polymers’ thermal, electrical, and mechanical properties, including rubbers [27,28,29,30]. The major advantage of graphene is to enhance the wear resistance of the composite, which has been extensively studied. When adding 3% expanded graphene and modified graphene to butadiene, Malas and his colleagues obtained a rubber composite with a significant enhancement of the wear resistance [31]. Graphene-added styrene-butadiene rubber was successfully fabricated with a 56% decrease in the wear rate of composites [32]. In another study, Wu’s group blended graphene with the mixed solution of butadiene and polymerized styrene-butadiene rubber, and the result showed a significant improvement in the wear resistance [33]. However, in order to improve the distribution and the interfacial connection of the graphene in the rubber, it is necessary to modify the graphene with functional groups to overcome Van der Waals’ force and the inert nature of the graphene in the rubber matrix.

Graphene can be modified using physical and/or chemical approaches. In the physical modification, the organic compounds or surfactants could be employed to enhance the dispersibility in solvents or water, respectively. Graphene’s surface was modified with dodecylamine and introduced into low-density polyethylene (LDPE); the resultant composite showed an enhanced conductivity and mechanical strength [34]. In another study, Mao’s group modified graphene with the butadiene styrene vinyl pyridine rubber latex to prevent the aggregation of graphene sheets and to act as an interfacial bridge between graphene and styrene-butadiene rubber [35]. The modified graphene sheets showed good distribution with no aggregation in the NR matrix. However, in this case, Mao added the modifier during the preparation of the composite, which might reduce the modification degree of the graphene surface. The majority of works aim to functionalize the graphene oxide before reducing it to obtain a functionalized graphene sheet. To the best of our knowledge, no work was conducted to modify pristine graphene nanoplatelets (GNPs) using dioctyl phthalate (DOP, a common plasticizer used in the rubber’s composition) to improve its dispersibility in the rubber matrix.

Herein, we proposed a simple approach to modify the pristine graphene nanoplatelets with dioctyl phthalate plasticizer. The modified GNPs were employed as an additive to improve the mechanical properties (especially abrasion resistance performance) of the NR. The dispersibility and nature of the modified graphene in the NR matrix were investigated. The effect of the modified GNPs’ contents on the mechanical properties (abrasion weight loss, tear strength, tensile strength, hardness, elongation at break…) of the DOP-modified GNPs/NR composite was studied and discussed in detail.

## 2. Materials and Methods

### 2.1. Materials

Graphene nanoplatelets were obtained from VNgraphene Joint Stock company, Vietnam. Sodium dodecyl sulphate (SDS), dioctyl phthalate (DOP) plasticizer, zinc oxide (ZnO), stearic acid, N-tertbutyl-2-benzothiazolsunfenamite, and sulphur were purchased from Xilong Chemical Co., Ltd. (Guangdong, China) Natural rubber (Vietnamese Standard Rubber-SVR 3L) was provided by Vietnam Rubber Latex Co., Ltd. (Ho Chi Minh City, Vietnam). All the chemicals were used as received without any further modification.

### 2.2. Modification of Graphene Nanoplatelets

Graphene nanoplatelets were modified with the DOP plasticizer using a combined high mixer and probe sonicator system as shown in Figure 1. This modifying process was self-developed by our group. In the typical procedure, the first 4% *w*/*w* GNPs powder was dispersed in water with the presence of SDS surfactant under probe-sonicating conditions for 24 h. Then, 15 g of the DOP plasticizer was introduced to the stable aqueous GNPs solution at a temperature of 80 °C for 2 h under high shear-mixing conditions. The obtained mixture was dried in the oven at 140 °C for 6 h to evaporate the water completely. The final product was stored at ambient conditions for further experiments and characterization.

### 2.3. Fabrication of DOP-Modified GNPs/NR Composite

The graphene/NR composite was fabricated using a facile melt–mixing approach with the compositions presented in Table 1. Firstly, the NR rubber was mixed with the modified GNPs and N-tertbutyl-2-benzothiazolsunfenamite accelerator to obtain mixture 1. Sulfur and the accelerator were then added to mixture 1 at 60 °C, at 50 rpm for 45 min to form mixture 2. Finally, mixture 2 was sheet-rolled by two rolling mill, followed by a pressed vulcanization process to form final products. The vulcanization process was carried out at a temperature of 150 °C under a pressure of 10 MPa for 20 min. The control sample was also fabricated using the same approach without the addition of the DOP-modified GNPs for comparative purposes.

For the measurement of the mechanical properties, the samples were prepared as follows: the specimens were prepared in a dumbbell shape with a thickness of 1.00 mm and the dimensions were cut following ASTMD-D412-D for tensile test and ASTMD-624-C tear test. Before testing, the specimens were placed into a climate-controlled box with a temperature of 25 °C and humidity of 60% for 24 h.

### 2.4. Characterizations

Scanning electron microscopy (SEM, Hitachi S-4600) was employed to observe the distribution of the modified GNPs in the NR matrix. The samples for SEM observation were coated with platinum as the conducting material. The morphologies of the samples were observed on the surface as well as on the cross-section sample. The crystallinity of the GNPs and the GNPs/NR composite were investigated using X-ray diffraction (XRD, X’Pert PRO PANalytical) with a 0.15405 nm Cu Kα source. The nature of the pristine GNPs and the modified GNPs in the NR matrix was investigated using Raman spectroscopy (Horiba XploRA Plus instrument, Kyoto, Japan). The abrasion resistance results were obtained with the rotating cylindrical drum device(GOTECH GTFO12D, Tokyo, Japan) with a drum diameter of 450 mm × 450 mm, pressing force of 2.5 N, and rotating cycles of 100. The INSTRON 5582 testing machine (Norwood, MA, USA) was employed to measure the tensile and tear strength of the samples. The elongation and break properties were determined according to the ASTM D412 standard on the STROGRAPH VG5E instrument (Fukuyama, Japan).

## 3. Results and Discussion

In order to form a good dispersion of graphene nanoplatelets (GNPs) into the rubber matrix, especially in the natural rubber, the GNPs should be modified with an appropriate compound that is compatible with the natural rubber. Dioctyl phthalate (DOP) is widely utilized as a plasticizer in the natural rubber formula. Additionally, DOP, as one of the organic compounds demonstrated good bonding with the graphene surface via π-π interaction [36,37]. Thus, DOP was selected as one of the ideal organic compounds to modify the graphene’s surface to improve the dispersibility of GNPs in the synthetic rubber matrix. Illustrated in Figure 1 is the modification process of the GNPs by DOPs. The initial GNPs powder is in black grey, dried, and has a porous appearance. After modification with the DOPs, the DOPs-modified GNPs exhibited a grey in color with a relatively sticky surface.

The surface morphologies of the graphene nanoplatelets before and after modification with the DOP were observed by scanning electron microscopy, and the results are shown in Figure 2. The pristine graphene nanoplatelets show a wrinkle and crumpled appearance with a lateral diameter ranging from 10–30 µm (provided by the supplier) (Figure 2a). The graphene sheets were vertically semi-transparent to the electron beam of the SEM instrument, which indicated that the GNPs consist of about 30–40 layers of single graphene sheets (equivalent to the thickness of approximately 10 to 20 nm) [38]. After modification with the dioctyl phthalate, the GNPs’ surface becomes smoother and thicker (Figure 2b). It can also be observed that the DOP-modified GNPs’ surface was not semi-transparent to the electron beam, indicating that the DOP successfully covered all the surfaces of GNPs.

Illustrated in Figure 3 are the electron microscopy images of the modified GNPs/NR rubber composite. It can be clearly seen that the modified GNPs are well-integrated into the NR matrix, and the GNPs’ surface is homogeneously covered by the NR layer (Figure 3a). The uniform distribution of the DOP-modified GNPs in the NR matrix was also confirmed by the cross-section SEM image as shown in Figure 3b. This result indicates that the modification of the graphene with the DOP plasticizer significantly improves the compatibility of the graphene in the NR rubber matrix, and as a result, enhances the properties of the rubber.

The crystallinity of the modified GNPs and the modified GNPs/NR composite was investigated by X-ray diffraction, as shown in Figure 4. As shown in Appendix A, the XRD pattern shows the amorphous nature of the dioctyl phthalate and the GNPs are of graphitic nature with a strong diffraction peak at around 26.6°. The XRD pattern of the modified GNPs consists of the broad peak at 26.6° with significantly decreased peak intensity, which is typical for the characteristic peak of the graphitic compound in the form of graphene multilayers [39]. When the modified GNPs were integrated into the became matrix, the diffraction peak of the GNPs was still observed; however, the peak becomes less sharp with lower intensity and shifted to 26.3°. This demonstrates the good distribution of the modified GNPs in the NR matrix. Additionally, the width of the characteristic peak of the GNPs in the NR matrix was wider than that of the modified GNPs, indicating the modified GNPs were uniformly covered with rubber matrix.

The Raman spectroscopy analysis excited at 663 nm was further employed to investigate the nature of the pristine GNPs and the modified GNPs in the NR matrix (Figure 5). The GNPs’ Raman spectrum shows two characteristic peaks at around 1340 and 1585 cm^−1^ corresponding to the D and G bands of the defects in the carbon network and sp^2^ C-H bonds in the graphene structure, respectively [40]. The D band peak is significantly broader and lower than the G band, indicating that the GNPs have few defects and low oxygen-containing groups on the surface. For the Raman spectrum of the GNPs-reinforced NR composite, the G band peak shifts to the wavelength of around 1620 cm^−1^, demonstrating the excellent interaction between the modified GNPs with the NR rubber. Furthermore, the intensity of the D and G bands of the GNPs in the NR matrix greatly decreased as a probe of uniformly covering the GNPs’ surface with the NR rubber. These results indicate that upon modification of the GNPs with the DOP compound, the distribution of the GNPs in the rubber significantly improves, and as a result, enhances the physic mechanical properties of the rubber composite.

The stress–strain curve of GNPs/NR with GNPs loadings of 0.1, 0.3, 0.5, and 0.7 phr are shown in Appendix A. The toughness of each sample was calculated by the area of the region under the stress–strain curve. The toughness of the GNPs/NR composites with GNPs loadings of 0.1, 0.3, 0.5, and 0.7 were determined to be 8875.571, 9935.675, 8913.518, and 8964.419, respectively. Materials are considered to be tough when they withstand with both high stresses and high strains. The GNPs/NR composite with a GNPs content of 0.3 phr showed the highest stress and strain; therefore, it had the best toughness among NR/GNPs composite. The higher contents of GNPs such as 0.5 and 0.7 phr led to a decline in toughness, which might be due to the agglomeration of GNPs in the rubber matrix.

The graphene nanoplatelets with high stiffness, good modulus, and substantial surface area have been extensively utilized as a filler to enhance the mechanical properties of the rubber [41]. One of the most studied applications of graphene used as filler in the rubber composite is to improve the abrasion resistance of the composite [16,42]. Figure 6 shows the abrasion weight loss of the rubber without GNPs and reinforced with the GNPs. Without the presence of the GNPs, the abrasion weight loss is determined to be approximately 0.3 g/cycle. Upon the addition of the modified GNPs, the abrasion weight loss is significantly decreased. The GNPs/NR composite weight only reduces by about 0.063 g for each cycle of the abrasive testing with the GNPs fraction of 0.1 phr. When 0.3 phr of the GNPs was added into the composite, only 0.03 g of the rubber composite was lost for each cycle of testing, which accounts for about a 1000% increase in the abrasion resistance of the NR rubber reinforced with the graphene. Further increases in the GNPs’ concentrations leads to the increase in the abrasion loss of the rubber composite with the value of about 0.06 and 0.074 g/cycle for 0.5 and 0.7 phr GNPs, respectively. The decrease in the abrasion resistance at a high GNPs’ contents is ascribed to the aggregation of the GNPs, leading to a low distribution degree of the graphene in the NR matrix. The abrasion loss at the loading of 0.3 phr was employed to compare the performance of the pristine GNPs and modified GNPs and the result is shown in Appendix A. It can be clearly seen that the abrasion loss for the GNPs/NR composite significantly increased after modification with the dioctyl phthalate. This is attributed to the improvement in the GNPs’ dispersion after DOP modification in the natural rubber’s matrix.

It has been well-demonstrated in the literature that the reinforcement of rubber with graphene also improves the other mechanical properties of rubber [43]. Figure 7 shows the tear strength property of the GNPs/NR composite. It can be seen that the introduction of graphene into the NR matrix enhances the tear strength of the composite. However, this improvement is negligible, accounting for only about 10% at the GNPs’ concentration of 0.3 phr. This is because this graphene content is considered to be the optimal concentration that induces the uniform distribution of the graphene in the NR matrix. At a high GNP content, the tear strength tends to decrease due to the aggregation of the graphene leading to the concentrated stress phenomenon that occurs when tear force is applied; as a result, the tear strength of the composite reduces. Appendix A shows the comparative results in tear strength of the GNPs/NR rubber reinforced with pristine GNPs and modified GNPs at the loading of 0.3 phr. It can be clearly seen the tear strength of the GNPs/NR composite slightly increased after modifying GNPs with the dioctyl phthalate.

The effect of the GNPs on the tensile strength of the NR composite was also investigated, and the results are shown in Figure 8. It can be clearly observed that the addition of the modified GNPs into the NR matrix negligibly affects the tensile strength of the composite. At a GNP concentration of 0.3 phr, only about 5% of the tensile strength increases compared to the control sample. At a further increase in the GNPs contents, the tensile strength decreases. This phenomenon might be due to the increase in the GNP contents, leading to the aggregation of graphene flakes; as a result, the graphene surface was not uniformly covered by the NR rubber. Appendix A shows the comparative results in tensile strength of the GNPs/NR rubber reinforced with pristine GNPs and modified GNPs at the loading of 0.3 phr. It can be clearly observed that the modification of GNPs with DOP decreases the tensile strength of the rubber composite compared to that of pristine GNPs.

The other mechanical properties of the modified GNPs/NR rubber composite were also studied, as shown in Figure 9. Generally, the addition of the GNPs into the NR matrix slightly reduces the elongation at break, residual elongation, and Shore A hardness of the composite. For the elongation at break, the value negligibly increases at the GNPs of 0.1 phr and decreases follows the further increase in GNPs content (Figure 9a). This is explained by the replacement of rubber molecules by the graphene nanoplatelets in the NR matrix, which hinders the stretching and sliding motion between polymer molecules; as a result, the elongation at the break of the composite decreases. The addition of the graphene also induces a slight reduction in the residual elongation and Shore A hardness of the resultant GNPs/NR composites, which might results from the plasticizing effect of the DOP (Figure 9b,c). Shore A hardness also decreases along with the increase in the modified GNPs contents (Figure 9c), demonstrating that the addition of the graphene makes the resultant rubber composite more flexible and durable, and this reduction might be also due to the presence of the DOP plasticizer. The addition of graphene negligibly reduces the glass transition temperature of the GNPs/NR composite from 61.8 to 61° with the GNPs content of 0.3 phr (Figure 9d). Appendix A exhibits the comparative results of the elongation at break, residual elongation, and Shore hardness of the GNPs/NR composite between the addition of pristine GNPs and modified GNPs. The results show that the use of pristine GNPs or modified GNPs negligibly affect to the elongation at break, residual elongation, and Shore hardness of the GNPs/NR composite.

## 4. Conclusions

In summary, DOP-modified graphene nanoplatelets were successfully fabricated and employed as a filler to significantly enhance the abrasion property of NR rubber. After modification with the DOP plasticizer, the GNPs showed good distribution in the NR matrix with a low degree of aggregation. The addition of the modified GNPs significantly increased the abrasion property of the NR rubber by 1000%, with only 0.3 phr of the GNPs in the composite. Higher GNPs concentrations in the NR composite showed the decreased improvement in abrasion resistance because of the graphene-aggregated phenomenon. While the addition of DOP-modified GNPs also enhanced the tear strength and tensile strength of the NR rubber, the elongation at break, residual elongation, and shore A hardness of the GNPs/NR composite slightly decreased. However, this reduction was negligible, making the DOP-modified GNPs an ideal additive with which to improve the abrasion property of NR rubber in a range of applications such as tires, conveyor belts, printing rolls, and shoes.

## Figures and Tables

**Figure 1 polymers-14-02541-f001:**
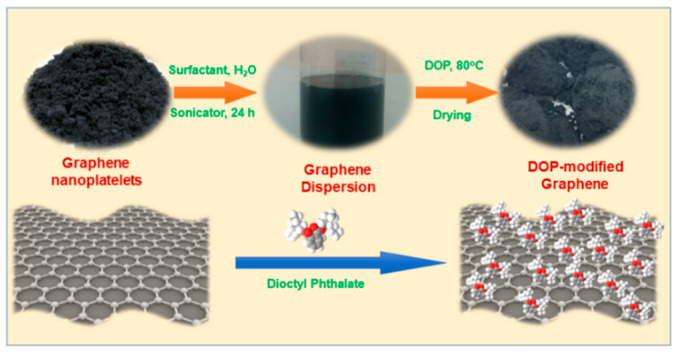
Fabricating process of the dioctyl phthalate-modified graphene nanoplatelets.

**Figure 2 polymers-14-02541-f002:**
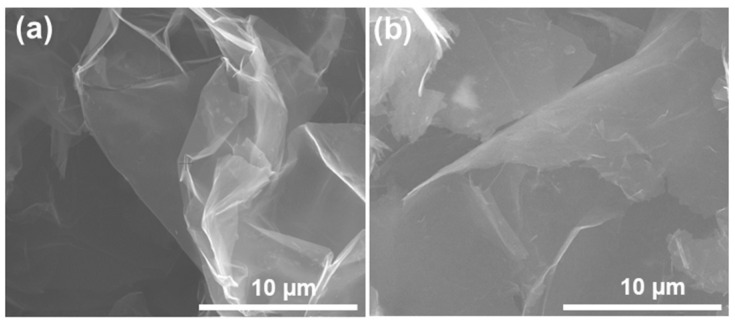
SEM images of the graphene nanoplatelets before (**a**) and after (**b**) modification with the DOP.

**Figure 3 polymers-14-02541-f003:**
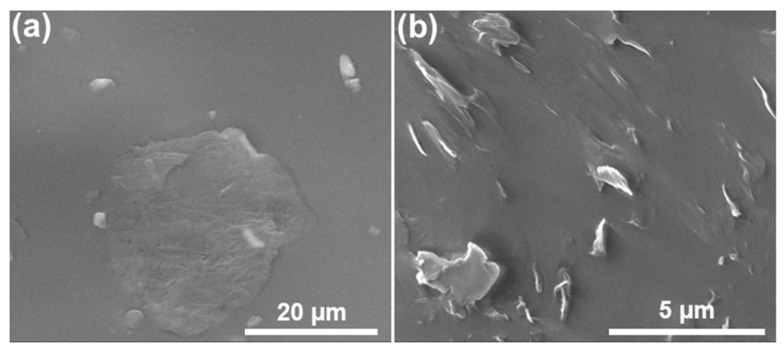
SEM images of the DOP-modified GNPs/NR rubber composite: (**a**) surface of the GNPs/NR rubber composite and (**b**) cross-section SEM image of the GNPs/NR rubber composite.

**Figure 4 polymers-14-02541-f004:**
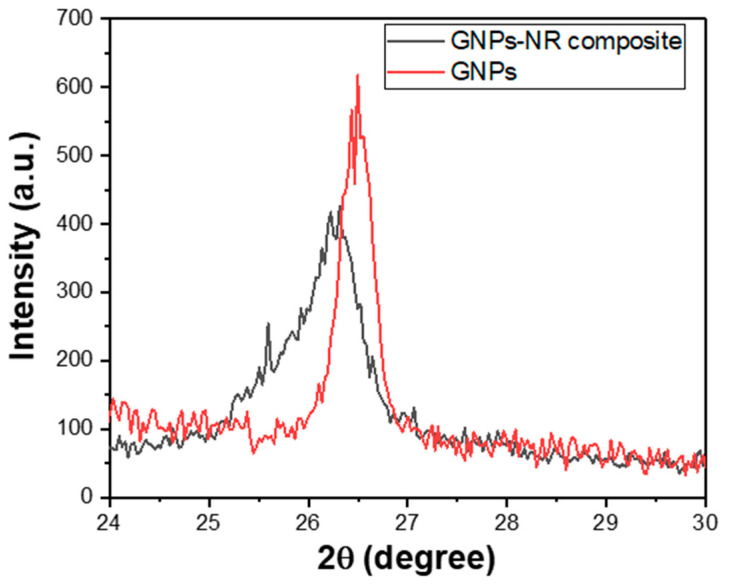
The XRD patterns of the modified GNPs (**red line**) and the modified GNPs/NR composite (**black line**).

**Figure 5 polymers-14-02541-f005:**
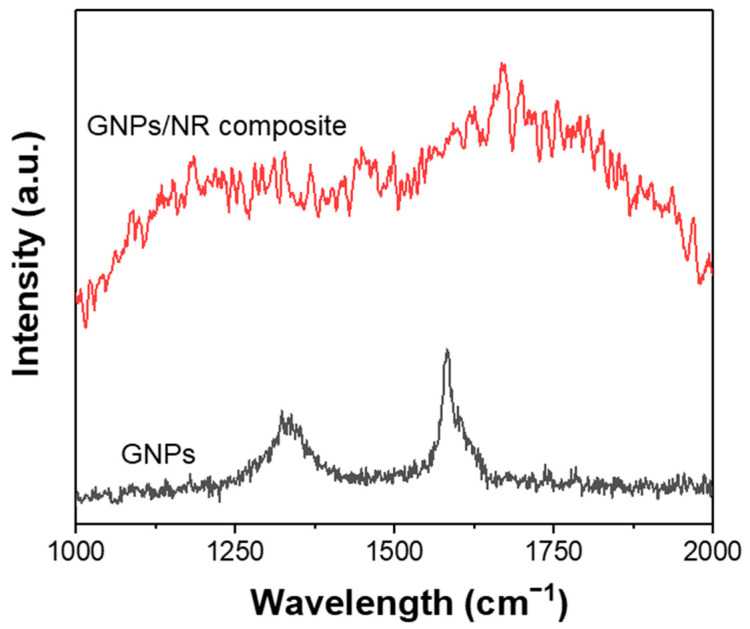
The Raman spectra of the modified GNPs (**black line**) and the modified GNPs/NR composite (**red line**).

**Figure 6 polymers-14-02541-f006:**
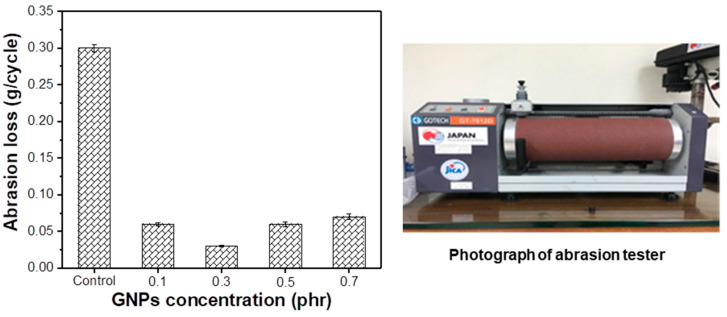
Abrasion loss of the control sample and the DOP-modified GNPs/NR rubber composites with various graphene contents.

**Figure 7 polymers-14-02541-f007:**
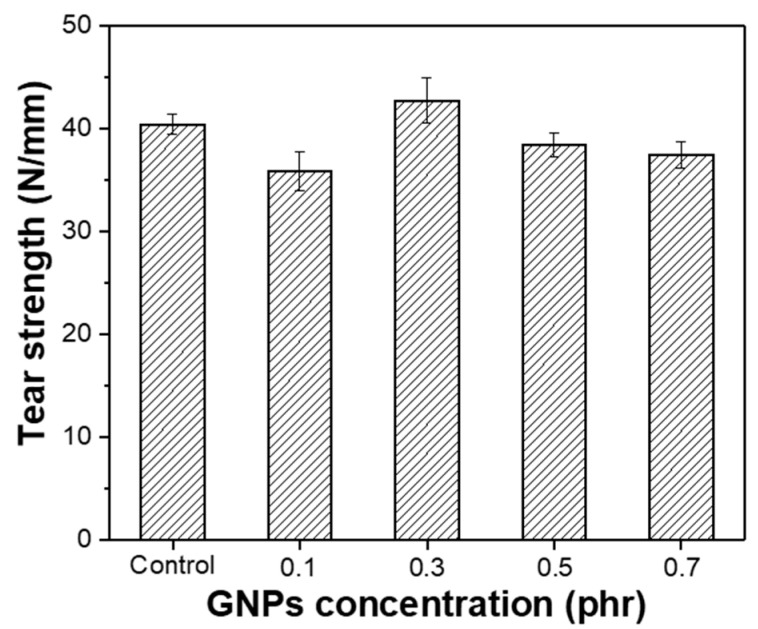
Tear strength of the control sample and the DOP-modified GNPs/NR rubber composites with various graphene contents.

**Figure 8 polymers-14-02541-f008:**
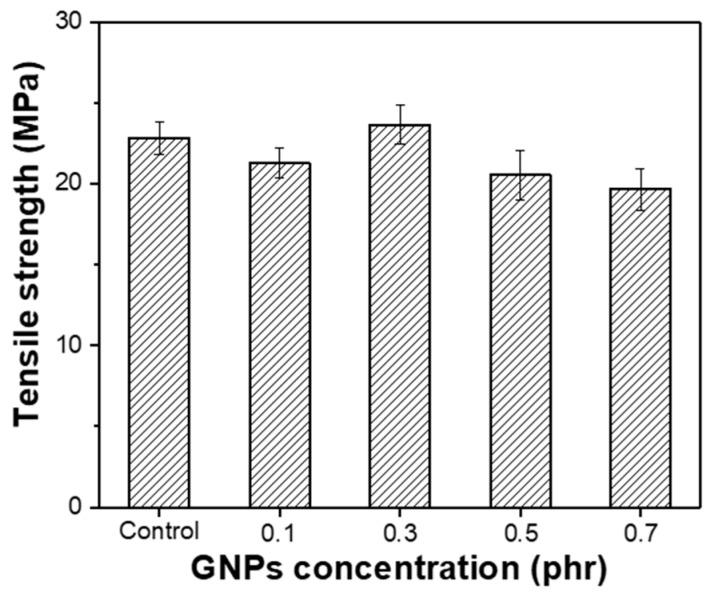
Tensile strength properties of the control sample and the DOP-modified GNPs/NR rubber composites with various graphene contents.

**Figure 9 polymers-14-02541-f009:**
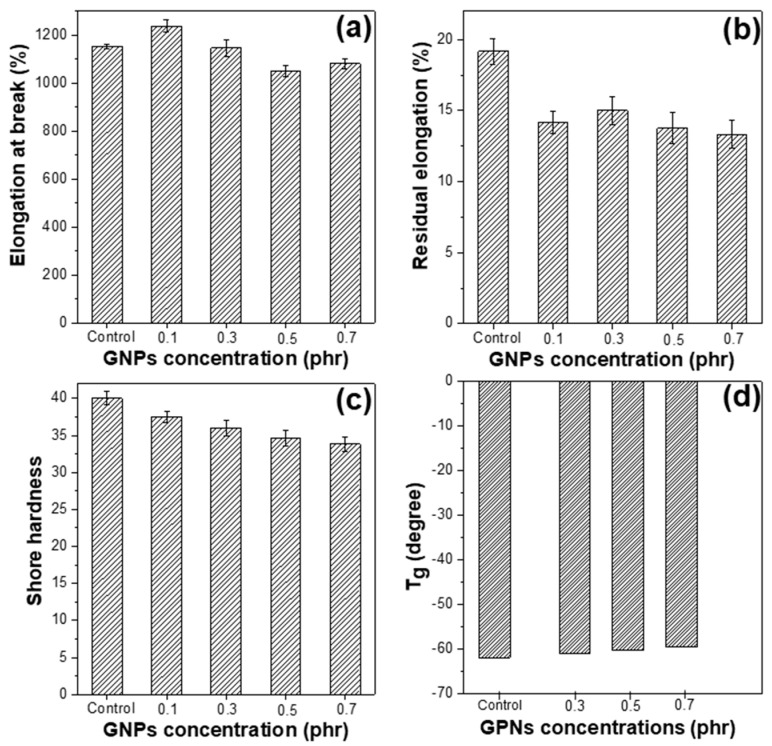
(**a**) Elongation at break, (**b**) Residual elongation, (**c**) Shore hardness, and (**d**) Glass transmission temperature of the control sample and the DOP-modified GNPs/NR rubber composites with various graphene contents.

**Table 1 polymers-14-02541-t001:** The composition of the DOP-modified GNPs/NR composite.

No.	Components	Ratio (phr *)
1	NR rubber	100
2	DOP-modified GNPs	0.1–0.7
3	Stearic acid	2
4	ZnO	5
5	TBBS accelerator	0.7
6	S	2.25

* phr: parts per hundred parts of rubber.

## Data Availability

Not applicable.

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
