# Peer review of "Dioctyl Phthalate-Modified Graphene Nanoplatelets: An Effective Additive for Enhanced Mechanical Properties of Natural Rubber"

_polymers, 2022, doi:10.3390/polym14132541_

Round 1

Reviewer 1 Report

Dear Authors,

In this research authors have investigated the reinforcing efficiency of DOP modified graphene in natural rubber composite. Authors have found good abrasion resistance properties by using the modified filler. This could be useful for making abrasion resistance rubber compounds. However, I have some recommendation that can be helpful for further improvements.

  1. What was the specific surface area of used graphene?
  2. Please mention the curing temperature and time of vulcanization and how it was obtained.
  3. Polarity of graphene might be lower than DOP modified graphene? I think the reinforcing effect (although it was not significant) of modified graphene may arouse from improved overall filler dispersion. Please provide XRD of pristine graphene and DOP modified graphene showing the change in the interlayer spacing.
  4. Please provide the abrasion resistance data using pristine graphene.
  5. Addition of solid material generally enhances the hardness of a rubber. Hence decreasing hardness quite anomalous. Please check the crosslink density of the rubber composites. Modified filler may have some negative effect to the cross-link density as this contains polar groups in their surface.
  6. These following papers could be helpful for improvements of this paper. a) Composites Communications, Volume 32, 101169 (2022) b) Polymer Bulletin, Volume 79, 2707–2724 (2022).

Author Response

June 12, 2022

Polymers

Re: Response on Manuscript ID polymers-1738022:

Thank you for considering our manuscript for publication in Polymers. We are very grateful for such a good news about the status of our manuscript. We have revised the manuscript to address the concerns of reviewers and Editors. Hope you find revised manuscript suitable for publication in Polymers.

We look forward to hearing from you in due course.

Yours sincerely

Dr. Duong Duc La

----------------------------------

Reviewer #1: 

  1. What was the specific surface area of used graphene?

Response: Thank you for the question. Since the specific surface area of a single layer graphene is about 2630 m2/g. The graphene nanoplatelets used in this work consist of 30 – 40 layers. Thus, the surface area of used graphene is estimated to be in range of 66 to 88 m2/g.

  1. Please mention the curing temperature and time of vulcanization and how it was obtained.

Response: Thank you very much for the valuable comment. The curing temperature and time of vulcanization were added. Please see the yellow highlight in the manuscript.

  1. Polarity of graphene might be lower than DOP modified graphene? I think the reinforcing effect (although it was not significant) of modified graphene may arouse from improved overall filler dispersion. Please provide XRD of pristine graphene and DOP modified graphene showing the change in the interlayer spacing.

Response: Thank you very much for the valuable comment. The XRD of the GNPs and DOP were added and discussed. Please see the yellow highlight in the manuscript.

  1. Please provide the abrasion resistance data using pristine graphene.

Response: Thank for the valuable comment. The abrasion resistance data of GNPs/NR using pristine graphene was added and discussed. Please see the yellow highlight in the manuscript.

  1. Addition of solid material generally enhances the hardness of a rubber. Hence decreasing hardness quite anomalous. Please check the crosslink density of the rubber composites. Modified filler may have some negative effect to the cross-link density as this contains polar groups in their surface.

Response: Thank you for the comment. The addition of the graphene also slightly reduces the residual elongation and Shore A hardness of the resultant GNPs/NR composites (Figure 9 b,c), may be due to plasticizing effect of DOP. This discussion was added. Please see the yellow highlight in the manuscript.

  1. These following papers could be helpful for improvements of this paper. a) Composites Communications, Volume 32, 101169 (2022) b) Polymer Bulletin, Volume 79, 2707–2724 (2022).

Response: Thank you very much for the valuable comment. These relevant works were added to improve this paper. Please see the yellow highlight in the manuscript.

Reviewer 2 Report

See document attached.

Author Response

June 12, 2022

Polymers

Re: Response on Manuscript ID polymers-1738022:

Thank you for considering our manuscript for publication in Polymers. We are very grateful for such a good news about the status of our manuscript. We have revised the manuscript to address the concerns of reviewers and Editors. Hope you find revised manuscript suitable for publication in Polymers.

We look forward to hearing from you in due course.

Yours sincerely

Dr. Duong Duc La

------------------------------

  1. I would recommend including in the Abstract some specific values (and/or conclusions) that were obtained for the properties described in that section (mechanical strength, abrasion loss...). This would be useful to put the reader in context about the improvements achieved and the information that can be found in the main text.

Response: Thank you very much for the valuable comment. The abstract was revised with addition of more specific values. Please see the yellow highlight in the manuscript.

  1. Based on the literature provided, the authors seem to have overlooked some very recent studies, published in high-impact editorials, related to the development of reinforced NR with different modified fillers. I suggest reviewing the following references and commenting on it to broaden the scope of the introduction (with a more current vision 1-3 years). I encourage authors to expand this list (or used one of their own).
  2. Composites Part C: Open Access, 2021, 5, 100133, DOI: 10.1016/j.jcomc.2021.100133
  3. Composites Science and Technology, 2020, 186, 107930, DOI: 10.1016/j.compscitech.2019.107930
  4. Composites Part B: Engineering, 2020, 200, 108346, DOI: 10.1016/j.compositesb.2020.108346
  5. ACS Omega, 2020, 5, 4, 1902-1910, DOI: 10.1021/acsomega.9b03516
  6. ACS Nano, 2020, 14, 3, 2788-2797, DOI: 10.1021/acsnano.9b09802
  7. Polymer, 2019, 161, 170-180, DOI: 10.1016/j.polymer.2018.12.014
  8. Nanomaterials, 2019, 9, 1, 12, DOI: 10.3390/nano9010012
  9. Polymer Testing, 2018, 67, 487-493, DOI: 10.1016/j.polymertesting.2018.03.032

Some questions could be answered: What has been done in the same area recently? Select a few interesting articles and commenting on it. What is new in this research compared with those 2 recent advances? What is the motivation for selecting this filler compared to others also used? Answering these questions in the Introduction could help improve it.

Response: Thank you very much for the recommendation. The introduction has been reviewed considering to the recommended questions. The authors have also carefully reviewed all the suggested works and added the recommended and related references to expand the knowledge in the field. Please see the yellow highlight in the manuscript.

  1. In the first paragraph, the authors mention as a disadvantage of conventional fillers (such as carbon black), the use of high contents that increase the production costs of the final elastomeric material. How does this relate to the use of special fillers as graphene? Do the low contents really compensate the costs? Carbon black is a cheap filler and above all very easy to produce, which continues to generate difficulties for its substitution

Response: Thank you very much for the valuable comment. Adding high proportion of the carbon black might lead to the reduction of the mechanical properties of the rubber. This claim was replaced for the high production cost of the elastomeric materials using carbon black. Please see the yellow highlight in the manuscript.

Moreover, Carbon black is one of typical filler for rubber composite industry, however it has to be used with the high content. Therefore, in some special applications carbon black might affect the mechanical properties of production and cause the high production weight. The graphene, which we used in this study, is the production from Vietnammese company thus the price of this graphene is not too expensive as the imported one. The low ammount of inorganic filler such as nano-graphene in production (0.3 phr) is not only does not affect much the production price but also increases some specific properties of production. On the orther hand, the application of high ammount of carbon black will cause evironment poluttion by dust during the manufaturing process. Meanwhile, the GNPs applied in this study was modified with DOP, one of the substrates in production receipt, did not affect the environment. With those facts, we saw the advantage of Vietnammese nanographene.

  1. The authors also state that: "recently, alternative nanoscale fillers have been widely studied and considered as an ideal replacement for conventional fillers in the rubber industry," please, explain the reasons.

Response: Thank you for the comment. The further explanation for the claim was added. Please see the yellow highlight in the manuscript.

  1. I recommend giving a little more detail of the techniques described in the Materials and methods section. For example:

- Is the process of modifying the GNPs with the DOP self-developed or based on previous literature reports? Specify.

Response: Thanks for the comment. The modifying process of GNPs with DOP was self-developed by our group. This was specified in the manuscript. Please see the yellow highlight in the manuscript.

- Under what conditions was the final product stored after modification?

Response: Thanks for the comment. The final product was stored at ambient conditions. This was specified. Please see the yellow highlight in the manuscript.

- Time and temperature of vulcanization? How was this time and temperature determined? Were curing curves performed?

Response: Thank you for the comment. The vulcanization process was implemented at temperature of 1500C under pressure of 10 MPa for 20 minutes. The vulcanization conditions were determined by using Rotorless Rheometer RLR-4-Toyoseiki (Japan) with curing curve.

- In what type of mixer were the mixtures prepared and for how long did the mixing take place?

Response: Thanks for the comment. The mixture were fabricated by hot mixer Labo Plastomill 4M150, the mixing process was divided in two steps:

+ Step 1: The NR rubber was mixed with the modified GNPs and N-tertbutyl-2-benzothiazolsunfenamite accelerator to obtain mixture 1. The hot mixing conditions were: 1100C, 50 rpm and 7 minutes.

+ Step 2: Sulfur and accelerator were then added to mixture 1 to form mixture 2. The hot mixing conditions were: 600C, 50 rpm and 5 minutes.

+ Finally, mixture 2 was sheet-rolled by two rolling mill, followed by a pressed vulcanization process to form final products.

- The meaning of “phr” (parts per hundred parts of rubber) is missing in the text.

Response: Thank you very much for the comment. The meaning of “phr” was added. Please see the yellow highlight in the manuscript.

- What type of specimens were used for the abrasion test? Dimensions and geometry? Were they directly vulcanized? Due to the difference in thickness compared to a sheet, was any modification in the vulcanization time required?

Response: Thank you very much for the valuable comment. The abrasion test was carried out according to ISO 4649 : 2002. The abrasion sample was in cylindrical shape with 16 mm in diameter and 6 mm in depth. By the higher thickness in comparion with sheet the vulcanization time was re-calculated by Vietnammese standard.

- Specify the type of specimen used for tensile and tear test. The way it is written is confusing.

Response: Thank you very much for the valuable comment. The dumbbell-shaped specimens were prepared by a gripping tool with the 1-mm-thick sample following ASTMD-D412-D. Likewise, the specimens for tear strength measurement were also prepared according to ASTMD-624-C. The samples were placed into stability conditions at room temperature for about 24 h before testing

Tensile sample

Tear test sample

- Specify details of sample preparation for scanning electron microscopy (SEM). Which section is observed? From cryogenic fracture or tensile test? Were the samples coated with a conductor?

Response: Thanks for the comment. The details of sample preparation for SEM observation was added. Please see the yellow highlight in the manuscript.

  1. On page 4, between the end of section 2 and the beginning of section 3, default content seems to have escaped from the journal template. Please check the submission carefully.

Response: Thank you very much for the valuable comment. The journal template between the end of section 2 and the beginning of section 3 was checked and corrected.

  1. I suggest changing the position of Figure 1 and moving it immediately after Modification of graphene nanoplatelets in the Materials and methods section.

Response: Thank you for the valuable comment. The Figure 1 was place right after Modification of graphene nanoplatelets in the Materials and methods section. Please see the yellow highlight in the manuscript.

  1. My major concern in this work is the absence of a standard formulation with unmodified GNPs. Consistently the authors claim improvements in properties compared to the unfilled formulation. This is not a fair comparison and avoids understand the usefulness of modification with DOP. The publication of this manuscript should be conditional on the inclusion of a base formulation, with unmodified GNPs, in all the characterization techniques used for the rubber composites (SEM, mechanical properties, abrasion and hardness). The authors can select the 0.3 phr content to optimize the workload, as this resulted in the best performance in the modified nanocomposites.

Response: Thank you very much for the valuable comment. The comparative results on the mechanical properties of the GNPs/NR composite between addition of pristine GNPs and modified GNPs. Please see the yellow highlight in the manuscript.

  1. It is consistent in the results to achieve a sweet spot in the properties at around 0.3 phr of the modified GNPs. From this content the performance starts to deteriorate. The authors justify this 3 fact on page 8, lines 245 - 247, where they state: "because of the weak interaction of graphenerubber and graphene-graphene, the micro-cracks develop quickly when the impact occurs, inducing the destruction of the GNPs/NR composite"; however, there are really no measurements that support this statement. Filler-filler and filler-rubber interactions are often indispensable parameters in the study of the reinforcing character of a new filler. Moreover, they are easily measurable through the Payne effect and the Mullins effect. I suggest the authors include these measurements to support these claims.

Response: Thank you very much indeed for the valuable comment. Since we don’t have enough time to do further experiment and measurements to clarify the claim "because of the weak interaction of graphenerubber and graphene-graphene, the micro-cracks develop quickly when the impact occurs, inducing the destruction of the GNPs/NR composite". So the remove this comment out of manuscript and tend to demonstrate this point in the future works.

  1. In Figure 9 the authors include a property called "Residual elongation", however, they do not give details on its determination. I can intuit that it is the remaining deformation after the tensile test and that it will be a measure of the elasticity of the material. If so, it would be expected that as additives outside the rubber vulcanization system are increased, the elastic character of vulcanized rubber would worsen; however, it seems that this is not the case. Why? Is this behavior supported by other reports in the literature? Including measurements of the crosslink density of the compounds or, better yet, of the compression set, would help to understand more easily the influence of the elastic and viscous phases of the viscoelastic material.

Response: Thank you very much for the valuable comment. The discussion in this point was correct to be more concise and logic. Please see the yellow highlight in the manuscript.

  1. In the title of Figure 9d, the authors report a "Transmission glass temperature". This is weird. Do they mean the glass transition temperature (Tg)?

Response: Thank you for the comment. This is glass transmission temperature. The error was correct in the title of Figure 9d, please see the yellow highlight in the manuscript.

  1. Adjust the Conclusions to the suggestions of other sections.

Response: Thank you for the comment. The conclusion was revised as recommendation.

  1. The authors close this section by stating that their development is ideal for NR in a "range of applications". Which ones? Please specify

Response: Thank you very much for the valuable comment. The range of application of NR with addition of modified GNPs was specified. Please see the yellow highlight in the manuscript.

Reviewer 3 Report

This manuscript is an experimental study on the dispersion of rubber matrix as the rubber is crosslinked in a standard way with sulfur chemistry. The authors utilized a common phthalate plasticizer to modify the graphene fillers and demonstrated a good degree of dispersion that was reflected in the mechanical properties of the prepared nanocomposites. Abrasion properties were also studied and very low abrasion loss values were measured. This reviewer had no major technical or scientific issues with this manuscript but is in the opinion that revisions are definitely needed to improve the quality and clarity of the work as follows:

  1. Please show stress-strain curves of the composites fabricated and also calculate the toughness of all samples from the area under the stress-strain curves and discuss the results.
  2. Can other rubbers like nitrile rubbers be effectively modified with this DOP-graphene system? Please comment.
  3. A relevant article on NR-graphene composite made with stretching properties should be included in the introduction section and briefly discussed: Journal of Industrial and Engineering Chemistry Volume 101, 25 September 2021, Pages 348-358.
  4. Please include a schematic or photograph of the abrasion measurement system in the manuscript.

Author Response

June 12, 2022

Polymers

Re: Response on Manuscript ID polymers-1738022:

Thank you for considering our manuscript for publication in Polymers. We are very grateful for such a good news about the status of our manuscript. We have revised the manuscript to address the concerns of reviewers and Editors. Hope you find revised manuscript suitable for publication in Polymers.

We look forward to hearing from you in due course.

Yours sincerely

Dr. Duong Duc La

--------------------------------

  1. Please show stress-strain curves of the composites fabricated and also calculate the toughness of all samples from the area under the stress-strain curves and discuss the results.

Response: Thank you very much for the valuable comment. The stress-strain curves of the GNPs/NR composite as well as the toughness of all samples under the stress-strain curves were added and discussed. Please see the yellow highlight in the manuscript.

  1. Can other rubbers like nitrile rubbers be effectively modified with this DOP-graphene system? Please comment.

Response: Thank you for the comment. In our other works with synthetic rubber lile nitrile, styrene butadiene rubber, butadiene rubber, the GNPs could be modified with DOP, Naphatalene oil, therefore it can enhance the properties of rubber matrix.

  1. A relevant article on NR-graphene composite made with stretching properties should be included in the introduction section and briefly discussed: Journal of Industrial and Engineering Chemistry Volume 101, 25 September 2021, Pages 348-358.

Response: Thank you for the recommendation. The recommended reference was added. Please the yellow highlight in the manuscript.

  1. Please include a schematic or photograph of the abrasion measurement system in the manuscript.

Response: Thank you very much for the comment. The photograph of the abrasion measurement system was added. Please see the yellow highlight in the manuscript.

Round 2

Reviewer 1 Report

Thank you for incorporate my comments and suggestion. All the best. 

Author Response

Thank you very much indeed for acceptance of our work to be published in Polymers

Reviewer 2 Report

Dear authors, thank you very much for considering all the reviewers´ suggestions. The manuscript has been improved considerably. Just two minor things:

- there is a typo on line 52

- regarding your answer to point 5, please include in the manuscript the mixing conditions (time, temperature, etc) 

Author Response

June 14, 2022

Polymers

Re: Response on Manuscript ID polymers-1738022:

Thank you for considering our manuscript for publication in Polymers. We are very grateful for such a good news about the status of our manuscript. We have revised the manuscript to address the concerns of reviewers and Editors. Hope you find revised manuscript suitable for publication in Polymers.

We look forward to hearing from you in due course.

Yours sincerely

Dr. Duong Duc La

-----------------------------------------------------------------

Reviewer #2: 

  1. There is a typo on line 52

Response: Thank you for the comment. The typo error was correct. Please see the yellow highlight in the manuscript.

  1. Regarding your answer to point 5, please include in the manuscript the mixing conditions (time, temperature, etc).

Response: Thank you very much for the valuable comment. The mixing conditions were added. Please see the yellow highlight in the manuscript.

Reviewer 3 Report

I found the manuscript revisions adequate and addressing all the comments raised. I can recommend its publication without further changes. 

Author Response

(The authors gave the same response as above.)
